# Time to revisit the skills and competencies required to work in rural general hospitals

**Cormac Doyle**[1]*, **Chris Isles**[2], **Pauline Wilson**[1]

**1** Department of Medicine, Gilbert Bain Hospital, Lerwick, Shetland, United Kingdom, **2** Department of Medicine, Dumfries and Galloway Royal Infirmary, Dumfries, United Kingdom

* Cormacdoyle@nhs.net

## Abstract

### Objectives

To determine the structure and demographic of medical teams working in Rural General Hospitals (RGHs) in Scotland, and to gain insight into their experiences and determine their opinions on a remote and rural medical training pathway.

### Design

Structured face-to-face interviews. Interviews were partially anonymised, and underwent thematic analysis.

### Setting

Medical departments of the six RGHs in Scotland 2018–2019.

### Participants

14 medical consultants and 23 junior doctors working in RGHs in Scotland.

Inclusion criteria: Present at time of site visit, medical consultant in an RGH or junior doctor working in an RGH who provides care for medical patients.

Exclusion criteria: Doctors on leave or off shift. Medical consultants with less than one month of experience in post. Non-medical specialty consultants e.g. surgical or anaesthetic consultants.

### Results

Of 21 consultant posts in the RGHs, only eight are filled with resident consultants, the remainder rely on locums. Consultants found working as generalists rewarding and challenging, and juniors found it to be a good training experience. Consultants feel little professional isolation due to modern connectivity. The majority of consultants (12/14) and all junior doctors favour a remote and rural medicine training pathway encompassing a mandatory paediatrics component, and feel this would help with consultant recruitment and retention.

**Data Availability Statement:** All demographic and quantitative data is available within the paper. The full transcripts of interviews cannot be made available to protect patient confidentiality, but are available on request to the NHS Shetland Clinical

Governance and Risk Team, via Fiona Morgan the NHS Shetland Clinical Audit Officer at Fionamorgan@nhs.net.

**Funding:** CD received funding from NHS Shetland via the Gilbert Bain Clinical Development Fellow Fund to cover travel costs to the RGHs. The study sponsor was not involved in study design, data interpretation, writing, or the decision to submit the article for publication. https://www.shb.scot.nhs.uk/hospital/gbh.asp The funders had no role in study design, data collection and analysis, decision to publish, or preparation of the manuscript.

**Competing interests:** All authors have completed the ICMJE uniform disclosure form at www.icmje.org/coi_disclosure.pdf and declare: no support from any organisation for the submitted work; no financial relationships with any organisations that might have an interest in the submitted work in the previous three years; no other relationships or activities that could appear to have influenced the submitted work. This does not alter our adherence to PLOS ONE policies on sharing data and materials.

## Conclusion

RGHs medical departments are reliant on locum consultants. The development of a remote and rural training medical training pathway is endorsed by the current medical teams of RGHs and has the potential to improve medical consultant staffing in RGHs.

## Introduction

There are six Rural General Hospitals (RGHs) serving Scotland's most remote and rural (R&R) populations (Fig 1) [1]. These are NHS hospitals in towns with small populations of over three thousand that require a consultant led service to meet their healthcare needs as they are more than two hours away from larger urban centres [2]. RGHs serve populations of around 20,000–44,000 people (Table 1) [3]. They usually have an emergency department (ED), a single medical ward and a single surgical ward staffed by 8–12 junior doctors, and led by 3–4 substantive surgical, medical, and anaesthetic consultants. Due to the small size their teams must practice as generalists, rather than specialists, providing emergency care and managing chronic diseases to all patients in their locality, including children [4]. Each RGH's ED triages patients to medicine or surgery, they are then reviewed by a junior doctor, and a medical or surgical consultant is ultimately responsible for their care. In critically ill patient, an anaesthetic consultant is normally involved soon after their arrival in the ED. RGHs can access support from specialist centres by telephone or videolink with options for patient transfer when required. This arrangement is known as the 'Hub and Spoke model' with the specialist centre being the hub, and RGHs the spokes. Each of the areas covered by the RGHs have their own primary care team and a local or regional public health team.

Globally, a high proportion of the population live in rural settings [5], and Scotland is no exception with 17% of the population residing in rural areas [6]. Recruitment and retention of healthcare staff in R&R areas has always been challenge to healthcare equity, in the UK and internationally [7–9]. In Scotland this is reflected with high turnover of healthcare staff and a reliance on locums in RGHs [10].

Research into recruitment and retention of healthcare to rural areas has been carried out on regionally, nationally, internationally [11–14]. The rural pipeline model of recruitment and retention suggests that exposure of medical students and trainee doctors to rural settings (through personal connection or via training) increases the likelihood that they will both work and remain in rural areas [15–19].

The Scottish Government has tried to improve recruitment and retention of doctors to R&R areas, evidenced by numerous connected initiatives over the last two decades, however the continued reliance on locum consultants at RGHs suggests these have not be as successful as intended [1, 4, 20]. Currently, only two of the five Scottish medical schools having formal links with RGHs [21, 22]. Moreover, only a minority of Scotland's junior doctors are exposed to RGHs during training [23]. Students at the remaining Scottish medical schools have created "Remote and Rural Medicine" Facebook groups suggesting an unmet interest [24–26]. Some Scottish remote and rural training pathways, in surgery and general practice (family medicine), have been developed in recent years which has been shown to impacted recruitment positively in some areas [27–31]. Currently, there is no R&R medical training pathway in Scotland to train hospital generalists.

Other nations facing the same challenge of ensuring healthcare equity across R&R areas, have been more successful. Canada and Australia each have national faculties of remote and

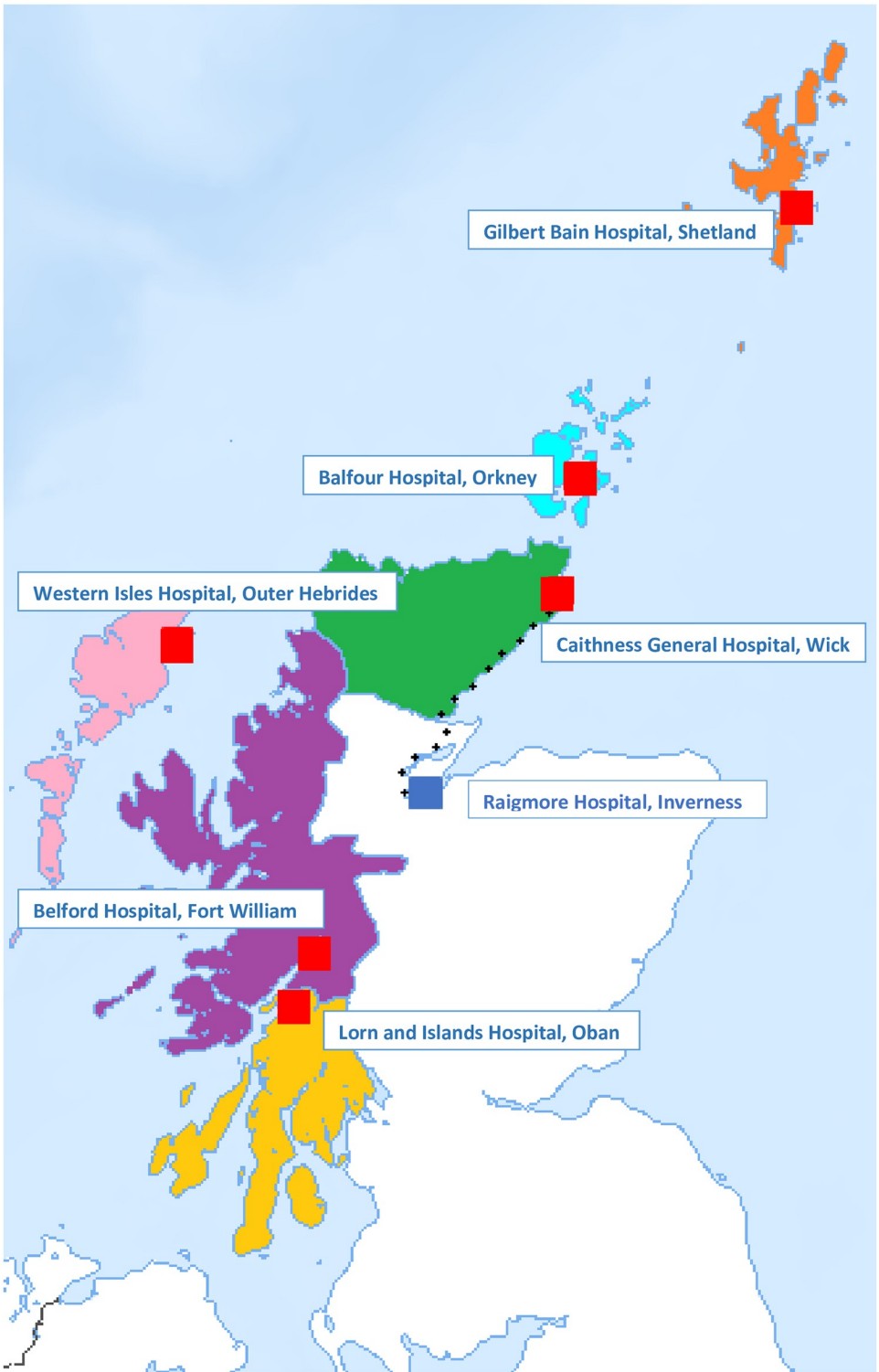

**Fig 1. Rural general hospitals, and health boards.** (The dotted line shows the 103 mile drive between Raigmore and Caithness General) Source: USGS National Map Viewer.

**Table 1. Populations served by RGHs in Scotland [3].**

| NHS Scotland Health Board | Population Number |
| --- | --- |
| NHS Shetland | 22,990 |
| NHS Orkney | 22,190 |
| NHS Western Isles | 26,830 |
| NHS Highland—Caithness[1] | 25,807 |
| NHS Highland—Fort William | 22,500 |
| NHS Highland–Oban[2] | 44,000 |

[1] Personal Communication via email with Ross MacKenzie NHS Highland Area Manager West, 2019.

[2] Personal Communication via email with Caroline Henderson Local Area Manager, Lorn & Island Hospital, 2019.

rural medicine that advocate for recruitment from rural communities, provide training courses for R&R healthcare staff and undertake research into remote and rural healthcare [32, 33]. Australia has an academic open source journal "Rural and Remote Health" [34]. The USA and New Zealand also have R&R healthcare training courses [35–37].

It is against this background, namely the difficulty in recruitment and retention to medical teams at RGHs and the absence of a R&R medicine training pathway, that we have chosen to study the structure and demographic of medical teams working in RGHs, gain insight into their experience of their role, and determine their opinions on a R&R medical training pathway.

## Methods

We designed questionnaires to be delivered to medical consultants and junior doctors working in each of the six RGHs in Scotland. Their aim was to gather information on demographic, previous training, experience of their role and ideas surrounding a R&R medicine training pathway.

Responses were gathered using structured face-to-face interviews between the interviewer and participant only, utilising open-questions to maximise responses, with no limit on interview duration. Separate questionnaires were designed for the medical consultants and junior doctors, in order to reflect the different roles and tenure of the two groups. The term junior doctor is used to describe all doctors below consultant grade, whether in a training programme or not.

These questionnaires were reviewed by members of the Remote and Rural Steering Group of the Royal College of Physicians of Edinburgh (RCPE) to provide feedback on both structure and content before being piloted locally in Gilbert Bain Hospital, Shetland. The final questionnaires used for the study can be viewed in S1 Appendix.

One of us (CD) visited all six RGHs between October 2018 and February 2019. CD was employed as a junior doctor working in an RGH at the time of interview, introducing potential bias in the data collection. All participants were aware of this before interview, and the questionnaire was followed in each interview to mitigate this bias.

Visits were arranged in advance by email or telephone. As many medical doctors as possible were interviewed at each site. Medical doctors on leave or off shift were excluded. A few doctors (n = 2) chose not to participate. We only interviewed locum consultants if they had more than one month's experience working at R&R sites. Junior doctors were interviewed regardless of the team they belonged to as they all cross-covered medicine and surgery when working out of hours.

Each interviewee was given a "Participant Information Sheet" detailing the projects aims, and there we no repeat interviews conducted. Participants signed a consent form before interview, and interviews were recorded on a password encrypted recording device. Data was stored in accordance with the GDPR.

The narrative responses were recorded, transcribed and partially anonymised, retaining only the job role of the interviewee and their hospital, they were not returned to participants for comment or correction. Thematic analysis was then performed by author CD using the method of Braun and Clarke, with consultant responses and junior doctor responses analysed separately [38]. Participants were sent a pre-submission draft of the manuscript to provide feedback on findings.

## Ethical approval

The Health Research Authority Decision tool was used during the design of this research and determined that research approval was not needed. Additionally, contact was made with the NHS Grampian Quality improvement and Assurance team who advised that ethical approval was not required.

## Results

The six rural general hospitals each have funding for between 3 and 4 medical consultant (Table 2). Five of the 6 employ medical consultants who undertake all of their clinical work locally at their RGH, and live in the surrounding area. Failure to recruit to these posts is common, and vacancies are covered by locums. Shetland, Orkney, Fort William and Oban all have permanent medical consultants, but also depend on locum consultants (Table 2). The Western Isles hospital in the Outer Hebrides is staffed by 3 regular locum consultants, one of whom is always on site.

The Caithness General Hospital in Wick has adopted a rotational model of medical consultant staffing (Table 2). Five medical consultants split their clinical time between their specialist work at Raigmore Hospital in Inverness (the hub) and generalist work in Caithness (the spoke). Each undertakes four days on call every fifth week in Caithness while the rest of their clinical time is spent in Inverness where they live (Fig 1). Weekends and any vacancies in the medical consultant rota are covered by locums.

**Table 2. Medical teams and interviewees in rural general hospitals.**

| Hospital | Number of consultant posts | Resident Consultants | Number of consultants interviewed | Number of Junior Posts | Number of juniors interviewed |
|---|---|---|---|---|---|
| Gilbert Bain Hospital, Shetland | 4 | 2 | 3 | 12 | 4 |
| Balfour Hospital, Orkney | 3 | 1 | 3 | 8 | 2 |
| Belford Hospital, Fort William | 3 | 2 | 2 | 9 | 5 |
| Lorn and Islands Hospital, Oban | 4 | 3 | 2 | 10 | 4 |
| Western Isles Hospital, Outer Hebrides | 3 | 0[1] | 1 | 12 | 5 |
| Caithness General Hospital, Wick | 4 | 0[2] | 3 | 10 | 3 |
| TOTAL | 21 | 8 | 14 | 61 | 23 |

[1] Three regular locums, at least one of whom is always on site.

[2] Five medical consultants who rotate on a regular basis from Raigmore Hospital, Inverness.

**Table 3. Consultants interviewed.**

| Hospital and Interviewee | Previous R&R | GIM | Specialty | Consultancy (years) | Current post (years) |
|---|---|---|---|---|---|
| Shetland 1 | Y | Y | Diabetes | 19 | 5 |
| Shetland 2 | Y | N | Cardiology | 11 | 1 |
| Shetland 3 | Y | Y | R&R[1] | 14 | 14 |
| Orkney 1 | Y | Y | Acute | 5 | 1 |
| Orkney 2 | Y | Y | Respiratory | 13 | 4 |
| Orkney 3 | Y | Y | Nephrology | 21 | 4 |
| Fort William 1 | Y | Y | GP | 6 | 10[2] |
| Fort William 2 | Y | Y | Gastro | 26 | 26 |
| Oban 1 | Y | Y | Diabetes | 25 | 1 |
| Oban 2 | N | Y | Nephrology | 18 | 1 |
| Western Isles 1 | Y | Y | None | 9 | 4 |
| Caithness 1 | Y | Y | None | 20 | 15 |
| Caithness 2 | Y | Y | R&R[1] | 7 | 4 |
| Caithness 3 | Y | Y | Acute | 3 | 3 |

[1]Specialist interest in Remote and Rural. During their specialist training each of these consultants had an individual agreement with their deanery to undertake extra training outside the scope of their speciality (e.g. Psychiatry for 6 months) to allow them to practice more effectively as a generalist in the future.

[2]Underwent Caesar Route to specialist accreditation in General Internal Medicine, initially being under supervision in their role whilst gaining consultant qualification.

## Medical consultants

One of us (CD) interviewed 12 male and 2 female consultants. Two were under 40 years of age, five were 40–49 years of age, four were 50–59 years old and three were over 60 years of age. All but one had been raised in an R&R area and/or had previous experience working in a R&R area before taking up their posts. Only 2 of the 14 had spent their entire consultant career in a RGH. Thirteen of the 14 had trained in GIM, but only two had created an R&R training pathway for themselves. Their other specialist interests were diverse as shown in Table 3.

## Thematic analysis of consultant interviews

### Theme 1: Generalist medicine

Consultants said they enjoyed the generalist aspect of their clinical practice, managing patients presenting with a broad range of acute and chronic diseases. This breadth was colourfully defined by one consultant as "if you can't put a knife in it, and she isn't about to deliver it's medicine and you deal with it". Their role was often contrasted with previous roles held within their speciality to highlight dealing regularly with unfamiliarity. Whilst this was satisfying, it was also a source of stress and challenge, and required regular advice from colleagues elsewhere in other specialties. The importance of finding a balance between, "giving things a go" and awareness of ones limitations,

> "It is proper general medicine, its challenging and refreshing"

> *Consultant 2*

> "It makes you very humble when you are in a position when you don't know, and it makes you feel like a junior doctor again when you are talking to a colleague in another speciality who knows more than you"

> Consultant 4

Working in small regular teams was considered a positive aspect of the role, allowing the development of deeper bonds with local colleagues. Several consultants commented that as the majority of junior doctors they worked with were at an early stage in their career, and there are no "middle grades" (specialist trainees below consultant grade) it can create more work, particularly supervising procedures.

## Theme 2: Professional isolation

Consultants spoke of the ease with which technology connected them to specialists and other information sources. Two factors did however occasionally contribute to professional isolation: a lack of understanding from colleagues at hub centres about the role and challenges of RGHs (generally considered to be a rare occurrence) and difficulty attending educational meetings that related to their specialty. On call commitments are higher at RGHs than at larger centres, and the travel time to Continued Professional Development (CPD) events is longer, making it inherently difficult to attend. Several consultants commented, that a richness and enjoyment in CPDs is lost when doing them online rather than in person. Consultants who rotated between Caithness and Raigmore did not experience feelings of professional isolation.

> "A willingness to realise you are isolated, and a willingness to do something about it is important"
>
> Consultant 2

## Theme 3: Training pathways

Most consultants (12/14) felt there should be a R&R medicine training pathway though some were concerned that it might deter otherwise competent medical consultants from applying and/or limit career options for a new consultant. There was no clear consensus, however, on how such a R&R medicine training pathway might be configured. Two options were proposed:

1. Trainees undertake General Internal Medicine training with a specialist interest in Remote and Rural medicine.

2. A post Completion Certificate of Training (CCT) syllabus of R&R skills and competencies could be completed by individuals if not already achieved during training and not already covered by the skills and competencies of consultants already in post.

Training experiences that were cited as most useful were acute takes; seeing a breadth of different pathologies, being responsible for a large number of patients and leading a team. It was felt that it would not be possible to see the volume of patients required for adequate training at R&R sites and therefore that the majority of training would need to occur at Hub centres.

## Theme 4: Paediatrics

Provision of care for paediatric patients was a particular source of stress and concern for consultants working in RGHs, even with remote support from specialist paediatric units. This was because of limited exposure to paediatrics in medical training programmes but initial responsibility for paediatric care presenting to the emergency department. Often medical consultants had voluntarily undertaken paediatric resuscitation courses, and in emergencies the RGHs anaesthetic consultants usually assisted. Only the Western Isles hospital has locum consultant

paediatricians on site. All agreed that some form of training in paediatrics was an essential requirement for a remote and rural training pathway.

> "I am not a paediatrician, but I'm expected to be able to manage the first stages of a paediatric emergency"

> Consultant 5

## Junior doctors

We interviewed 12 male and 11 female junior doctors. Twenty one were 20–29 years of age, one was 30–39 years of age, and one was 40–49 years of age. Twelve of the juniors had been raised in an R&R area, enjoyed a previous R&R placement in medical school, or belonged to an R&R student society. Fifteen juniors were in training posts and eight were in locum posts. Seven trainees and seven locums applied specifically to work in an R&R area. Fifteen juniors said they were interested in working in R&R areas in the future (Table 4).

Each of the six rural general hospitals have funding for between 8 and 12 junior doctors and offer a mixture of training and non-training posts. The most senior of these are core trainees in medicine and surgery. There are no registrars at any of the six RGHs (Table 4). All

**Table 4. Junior doctors interviewed.**

| Hospital | Previous R&R | Stage of training[1] | Applied to work R&R | Future interest R&R |
|---|---|---|---|---|
| Shetland 2 | N | Locum FY2 (3) | Y | Y |
| Shetland 3 | N | Locum FY2 (3) | Y | N |
| Shetland 1 | Y | Locum FY2 (3) | Y | Y |
| Shetland 4 | Y | FY2 | N | N |
| Orkney 1 | Y | Locum FY2 (4) | Y | Y |
| Orkney 2 | Y | Locum FY2 (4) | Y | Y |
| Fort William 1 | N | FY1 | N | N |
| Fort William 2 | N | GPST1 | Y | Y |
| Fort William 3 | Y | FY1 | Y | Y |
| Fort William 4 | Y | Locum FY2 (3) | Y | N |
| Fort William 5 | N | FY2 | N | N |
| Oban 1 | Y | GPST1 | Y | Y |
| Oban 2 | Y | FY2 | Y | Y |
| Oban 3 | N | FY1 | N | N |
| Oban 4 | N | FY1 | Y | N |
| Western Isles 1 | Y | GPST1 | Y | Y |
| Western Isles 2 | N | FY2 | N | Y |
| Western Isles 3 | Y | FY2 | Y | Y |
| Western Isles 4 | N | CST1 | N | N |
| Western Isles 5 | Y | FY2 | N | Y |
| Caithness 1 | N | FY2 | N | N |
| Caithness 2 | N | Locum CMT2 | Y | N |
| Caithness 3 | Y | Locum FY2 (4) | N | Y |

R&R = Remote and Rural; FY1 = Foundation year 1; FY2 = Foundation Year 2.

GPST1 = General Practice Specialty Trainee 1; CST1 = Core Surgical Trainee 1.

[1]Number of years since qualification is denoted in brackets for junior doctors in locum posts.

junior doctors, except FY1s, contribute to the on call rota. The exception was the Western Isles where junior doctors do not undertake night shifts, which are provided by an A&E specialist, a paediatrician, and specialist nurses.

## Thematic analysis of junior doctor interviews

### Theme 1: Generalist medicine

Juniors also found working as a generalist a useful and enjoyable learning experience. They saw a breadth of pathology, grew in confidence as they learned how to manage patients at night with limited access to tests, and were "encouraged to think like registrars" as they discussed queries directly with consultants whenever needed. Some juniors missed having "middle grades" whom they could approach for advice as they had done in larger centres.

> "You get to do a lot more clinically, you don't get this hands on experience in a less remote setting"

*Junior Doctor 3*

Some juniors found working in the Emergency Department difficult. Of note they felt supported when dealing with major cases, as they would get consultant help, but several reported struggling with "minor injuries" due to lack of formal emergency department training.

> "I don't like dealing with minors in A&E, I'm unsure of what to do, but it can feel too small to ask for help so I have to spend a lot of time looking it up. . . . . . . . ... And I'm slow at suturing"

Junior 6

All of the juniors enjoyed working in small teams amongst a relatively small population, as they established better relationships with their colleagues and regular patients. The team sizes meant the rota was less flexible, which could lead to issues regarding leave and attending training courses.

> "You get to know you patient better and your colleagues better"

Junior Doctor 10

### Theme 2: Training pathways

All juniors believed there should be a training pathway for consultants in R&R medicine and that this would improve recruitment. Their view was that the training pathway should follow general medical training and that R&R medicine should be an independent specialism. They felt the pathway should be "more general" than other medical training pathways, but weren't sure how this might be achieved and other than paediatric training, no specific examples were given of what should be in the pathway. This may reflect that all juniors interviewed had not undertaken any speciality training as they were in the first few years of their career.

### Theme 3: Paediatrics

Juniors found seeing paediatric patients stressful, normally due to limited paediatric training. Training in paediatrics was mentioned by each junior as something that would be a vital part of a remote and rural training pathway.

"I find seeing the paediatric cases in A&E difficult"

Junior doctor 11

## Discussion

The main findings of our study are: that failure to recruit to medical consultant posts in Scottish RGHs is common with frequent use of locums to cover vacancies; that medical consultants' specialist interests are diverse; and that consultants enjoy the generalist aspect of their clinical practice but do occasionally experience feelings of professional isolation. Most felt there should be a Scottish R&R medicine training pathway and that some form of training in paediatrics was an essential requirement, though there was no clear consensus on how such a training pathway might be configured. Juniors also found working as a generalist a useful learning experience and grew in confidence as they learned how to manage patients at night with limited access to tests. All agreed that training in paediatrics should be a vital part of a remote and rural training pathway.

RGHs currently offer one of two models of consultant led care. Five hospitals (Shetland, Orkney, Oban, Fort William and Western Isles) have attempted to staff their medical units with consultants who live locally and embed themselves in their local communities. This has only been partially successful as evidenced by the fact that all five currently require locum consultants to fill gaps. One RGH (Wick) rotates permanent medical consultant from the hub in Inverness to the spoke in Caithness on a weekly basis but still relies on locum consultant cover at weekends. It is our view that what may work well in one hospital may not be suitable or appropriate in another. Rotating between Inverness and Caithness may remove feelings of professional isolation, but may lead to challenges in continuity of care and may be a less than ideal solution for the three island RGHs.

Health care equity is a guiding principle of the NHS [39]. The delivery of generalist secondary care medical services to remote and rural areas of Scotland needs initiatives that work. Central to this discussion must be the creation of a training pathway(s) that encourages doctors to consider a career in a RGH. Thematic analysis suggested two possibilities: either that trainees undertake Remote and Rural medicine as a specialist interest, or a post CCT syllabus of R&R skills and competencies mapped to the needs of individuals and the skill set of the team they are joining. While we recognise that there will always be social and demographic factors such as distance from family and friends that deter some doctors from working in RGHs we believe that the creation of a medical remote and rural training pathway(s) is a pragmatic solution to the workforce challenges faced by RGHs [17, 40–42]. Recruitment to this pathway could focus on doctors with remote and rural exposure, as is the case for no fewer than 13 of 14 consultants interviewed, supporting the rural pipeline model of recruitment.

We are not the first to suggest that remote and rural medicine should have its own training pathway in Scotland. Wilson and McHardy, in their review of remote and rural training in Scotland in 2001–2, noted the tendency of hospital training programmes to prepare sub-specialists for secondary and specialist care centres, rather than training generalists who might work in remote and rural areas where a wider repertoire of less specialised skills and roles are

necessary. They went on to list the skills and competencies that might be required of a training programme for remote and rural medical consultants [43].

Now may be the time to form this training pathway. The established remote and rural training pathways in general practice (family medicine) and surgery provide local examples, and give the potential for collaboration [28, 31, 44]. The Scottish Executive, UK government, and the UK General Medical Council have all recognised the issue of staffing R&R areas and made recommendations about the training needs of generalists [45–47]. while other nations provide researched models of R&R training pathways and their impact on recruitment, with evidence of collaboration that Scotland could emulate [48–55].

In conclusion, Rural General Hospitals are vital to the provision of good healthcare across Scotland. Their medical units evidently need new approaches to recruitment and retention of consultant staff. The development of a remote and rural medicine training pathway(s) would aid recruitment by creating a clear route for interested junior doctors and consultants to follow, and would prepare them to meet the needs of the people who live in remote and rural areas. This would follow the trajectory taken by other nations with more robust remote and rural healthcare systems [54, 55]. Much has changed since Wilson and McHardy published their 2004 recommendation [43]. All six of the RGHs now have their own CT scanners, and telemedine and guidelines are the rule rather than the exception, but the fundamental issue of the training of generalist medical consultants remains. We believe it is time to follow the surgeons and general practitioners, and our international colleagues, and clarify the skills and competencies required of a generalist medical consultant. This is a necessary step to ensure healthcare equity throughout Scotland.

## Supporting information

**S1 Appendix. Consultant and trainee questionnaires.**
(DOCX)

**S1 File.**
(PDF)

## Acknowledgments

We thank all the physicians who took part in the study. We also thank the members of the Royal College of Physicians of Edinburgh's Remote and Rural Steering group for reviewing the questionnaires.

## Author Contributions

**Conceptualization:** Cormac Doyle, Chris Isles, Pauline Wilson.

**Data curation:** Cormac Doyle.

**Formal analysis:** Cormac Doyle.

**Funding acquisition:** Cormac Doyle, Pauline Wilson.

**Investigation:** Cormac Doyle.

**Methodology:** Cormac Doyle.

**Project administration:** Cormac Doyle.

**Supervision:** Chris Isles, Pauline Wilson.

**Writing – original draft:** Cormac Doyle, Chris Isles.

**Writing – review & editing:** Cormac Doyle, Chris Isles, Pauline Wilson.

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
