## [Decision Letter · Decision Letter 0]

4 May 2020

PONE-D-20-06347

Time to revisit the skills and competencies required to work in Rural General Hospitals

PLOS ONE

Dear Dr Doyle,

Thank you for submitting your manuscript to PLOS ONE. After careful consideration, we feel that it has merit but does not fully meet PLOS ONE’s publication criteria as it currently stands. Therefore, we invite you to submit a revised version of the manuscript that addresses the points raised during the review process.

The manuscript addresses an important topic for rural and remote health care not just in Scotland but in many countries. Both reviewers have highlighted the importance of situating the work in a broader context, particularly given the international audience of the journal. In addition reviewer 2 has provide some suggestions for enriching the analysis of the results and I invite you to consider these comments in revising the work.

We would appreciate receiving your revised manuscript by Jun 18 2020 11:59PM. To enhance the reproducibility of your results, we recommend that if applicable you deposit your laboratory protocols in protocols.io, where a protocol can be assigned its own identifier (DOI) such that it can be cited independently in the future. For instructions see: http://journals.plos.org/plosone/s/submission-guidelines#loc-laboratory-protocols

We look forward to receiving your revised manuscript.

Kind regards,

Jenny Wilkinson, PhD

Academic Editor

PLOS ONE

Journal Requirements:

Additional Editor Comments (if provided):

Reviewers' comments:

Reviewer's Responses to Questions

**Comments to the Author**

1. Is the manuscript technically sound, and do the data support the conclusions?

Reviewer #1: Yes

Reviewer #2: No

2. Has the statistical analysis been performed appropriately and rigorously? 

Reviewer #1: N/A

Reviewer #2: N/A

3. Have the authors made all data underlying the findings in their manuscript fully available?

Reviewer #1: Yes

Reviewer #2: Yes

4. Is the manuscript presented in an intelligible fashion and written in standard English?

Reviewer #1: Yes

Reviewer #2: Yes

5. Review Comments to the Author

Reviewer #1: This study presents original research results that do not appear to have been published elsewhere. The analysis is appropriate for this qualitative study and is described in sufficient detail. Conclusions are presented appropriately and supported by the data. The article is intelligibly written in clear English. The appropriate ethics decision tool and consultation determined that research ethics approval the was not needed. The article appears to adheres to appropriate reporting guidelines of the standard for data availability. The abstract concisely describes the paper.

The introduction is clear, however for an international audience it would be helpful to provide more detail about the context, specifically what other doctors are part of the healthcare team at the rural general hospitals that were studied. The literature review is appropriate for this Scottish paper, but for an international audience could be expanded to include broader references to rural generalist practice in rural general hospitals in other comparable international settings.

The method is clearly described including identification of potential bias.

The results are clearly presented and form the basis of an excellent discussion

Reviewer #2: Thank you for the opportunity to review this paper. It is on the important topic of equity between rural/remote and urban populations with respect to medical care in Scotland. However it shows remarkably little insight into the large volume of publications from elsewhere in the world around this topic. In Australia, the a College of Rural and Remote Medicine (ACRRM) which does exactly as they have recommended has been in existence for more than two decades. Remote medical schools have existed in Canada and the US for 30 years with the mandate of producing a rural medical workforce, with evidence that they do so. The Rural and Remote Health journal, also based in Australia, has a list of relevant publications on this topic – plus more that could be found in a medline search. The present paper would be greatly enriched by considering this literature, and how care is provided elsewhere in the world where are populations are thousands, not hundreds, of kilometres far from city centres.

The method was suitable, and the themes not unexpected, however, the data appear not to be as rich as would be expected. With respect to the consultants, how does a general physician deal with road accidents requiring emergency surgery for example? How do they manage critical care? How do they ensure appropriate immunisation schedules for infants? Do they provide obstetric care, particularly for obstetric emergencies? These scenarios are surely commonplace in remote practice, and one would assume that some depth of what “generalism” meant should have come up in the interviews. For example how does an endocrinologist with 19 years’ experience deal with stroke in an elderly person, or a 21-year qualified Neurologist deal with sceptic diabetic ulcers?

The same could be said of the junior doctor interviews. If they are managing nights in a hospital without resident consultants, who do they call when the case management is entirely out of their competency? How to they feel about this? Is that why they think a new specialisation is needed? What do they think should be in it? I doubt that paediatrics difficulties is all they encounter!

I would suggest that the authors read the literature, and enlarge the scope of their interviews, and resubmit a new manuscript. I highly recommend that they do so, because they are correct in thinking this is an important area to research and publish in with insights into the local conditions pertaining in Scotland.

6. PLOS authors have the option to publish the peer review history of their article (what does this mean?). If published, this will include your full peer review and any attached files.

Reviewer #1: Yes: Professor James Rourke, Honorary Research Professor and former Dean of Medicine, Memorial University of Newfoundland, Society of Rural Physicians of Canada co-chair of Rural Road Map Implementation Committee

Reviewer #2: No

---

## [Author Response · Author response to Decision Letter 0]

19 Aug 2020

Dear Reviewers,

Thank you for taking the time to read and review our manuscript. We have found your comments extremely helpful and feel our revised manuscript is a much more interesting piece of work as a result. 

There have been two major changes to the paper. The first is to make the paper more readable to an international audience by expanding our description of Scottish Rural General Hospitals (RGHs) and discussing remote and rural healthcare internationally. 

The second major change is clarification of our exclusion criteria and expanding our thematic analysis section to give more detail from interviewees.

Below is are more detailed responses to specific comments (reviewers comments are in bold italics)

The introduction is clear, however for an international audience it would be helpful to provide more detail about the context, specifically what other doctors are part of the healthcare team at the rural general hospitals that were studied. 

In our introduction, we have expanded the first paragraph describing the Rural General Hospitals (RGHs). We have added detail about how patients are managed in the emergency department, the role of anaesthetic and surgical consultants, and the wider healthcare infrastructure in the areas covered by RGHs (primary care and public health). We feel these changes have added clarity, particularly for international readers.

The literature review is appropriate for this Scottish paper, but for an international audience could be expanded to include broader references to rural generalist practice in rural general hospitals in other comparable international settings.

However it shows remarkably little insight into the large volume of publications from elsewhere in the world around this topic. …The present paper would be greatly enriched by considering this literature, and how care is provided elsewhere in the world where are populations are thousands, not hundreds, of kilometres far from city centres.

We have made changes throughout the paper to discuss remote and rural healthcare internationally, and feel this has enriched the paper. In the introduction, we discuss how Scotland is not unique in needing to provide healthcare in remote and rural settings to ensure healthcare equity, and describe how other nations have succeeded. In the discussion, we re-iterate that other nations provide an example of remote and rural healthcare that Scotland can follow, and cite researched examples in Australia and Canada showing the impact of training on recruitment and retention.

Throughout the manuscript, we have tried to be more mindful towards international readers, who are unlikely to be familiar with the idiosyncrasies of the NHS in remote and rural Scotland. We have summarised a paragraph in the introduction that mentioned numerous Scottish government policy papers, and throughout have tried to use healthcare related terms that will be understood internationally.

The method was suitable, and the themes not unexpected, however, the data appear not to be as rich as would be expected. With respect to the consultants, how does a general physician deal with road accidents requiring emergency surgery for example? How do they manage critical care? How do they ensure appropriate immunisation schedules for infants? Do they provide obstetric care, particularly for obstetric emergencies? These scenarios are surely commonplace in remote practice, and one would assume that some depth of what “generalism” meant should have come up in the interviews. For example how does an endocrinologist with 19 years’ experience deal with stroke in an elderly person, or a 21-year qualified Neurologist deal with sceptic diabetic ulcers?

The same could be said of the junior doctor interviews. If they are managing nights in a hospital without resident consultants, who do they call when the case management is entirely out of their competency? How to they feel about this? Is that why they think a new specialisation is needed? What do they think should be in it? I doubt that paediatrics difficulties is all they encounter!

This was a helpful comment, as our original manuscript did not include much detail on how the RGHs operate, particularly between the different teams of doctors. We have added to the introduction about the management of major cases, and that there are primary care and public health teams covering these areas who co-ordinate immunisations. We have also tried to clarify that we only interviewed medical consultants, not surgical or anaesthetic consultants, and have explicitly added this to the exclusion criteria. Medical consultants would have little involvement in obstetric care and road accidents at RGHs, the surgical and anaesthetic teams would manage this.

We have expanded the thematic analysis sections. We have added detail about how some consultants described “generalism” as a concept. Some of the lack of richness, may be due to the questions we asked. The consultants did not go into details of cases outside their specialties, but discussed this concept generally. 

For the junior doctor thematic analysis, we have expanded the difficulties they encounter, namely that they struggle with “minor injuries” in the ED due to lack of previous experience, and with having no “middle grade” doctors to go to for advice. 

While all of the juniors felt there should be a remote and rural medical training pathway, none gave details except that paediatrics should be included. This may reflect that they are all within the first few years of their career, and have not undergone any specialism training. Additionally, the juniors interviewed have only worked in RGHs for a number of months, and may not have reflected as deeply as the consultants on the need for such a pathway. 

Thank you for your consideration, we hope this is a clear response to your helpful comments.

Sincerely,

Authors of manuscript

---

## [Decision Letter · Decision Letter 1]

23 Sep 2020

Time to revisit the skills and competencies required to work in Rural General Hospitals

PONE-D-20-06347R1

Dear Dr. Doyle,

We’re pleased to inform you that your manuscript has been judged scientifically suitable for publication and will be formally accepted for publication once it meets all outstanding technical requirements.

Kind regards,

Jenny Wilkinson, PhD

Academic Editor

PLOS ONE

Additional Editor Comments (optional):

Thank you for your responses to reviewer comments and manuscript revisions. These have satisfactorily addressed the comments.

Reviewers' comments:

Reviewer's Responses to Questions

**Comments to the Author**

1. If the authors have adequately addressed your comments raised in a previous round of review and you feel that this manuscript is now acceptable for publication, you may indicate that here to bypass the “Comments to the Author” section, enter your conflict of interest statement in the “Confidential to Editor” section, and submit your "Accept" recommendation.

Reviewer #1: All comments have been addressed

2. Is the manuscript technically sound, and do the data support the conclusions?

Reviewer #1: (No Response)

3. Has the statistical analysis been performed appropriately and rigorously? 

Reviewer #1: (No Response)

4. Have the authors made all data underlying the findings in their manuscript fully available?

Reviewer #1: (No Response)

5. Is the manuscript presented in an intelligible fashion and written in standard English?

Reviewer #1: (No Response)

6. Review Comments to the Author

Reviewer #1: (No Response)

7. PLOS authors have the option to publish the peer review history of their article (what does this mean?). If published, this will include your full peer review and any attached files.

Reviewer #1: **Yes: **Dr James Rourke, Professor Emeritus and former Dean of Medicine, Memorial University of Newfoundland, St. John's, NL, Canada

---

## [Editor Report · Acceptance letter]

29 Sep 2020

PONE-D-20-06347R1 

Time to revisit the skills and competencies required to work in Rural General Hospitals 

Dear Dr. Doyle:

I'm pleased to inform you that your manuscript has been deemed suitable for publication in PLOS ONE. Congratulations! Your manuscript is now with our production department. 

Kind regards, 

on behalf of

Dr Jenny Wilkinson 

Academic Editor

PLOS ONE